# Optical *N*-invariant of graphene's topological viscous Hall fluid

Todd Van Mechelen [1], Wenbo Sun [1] & Zubin Jacob [1✉]

Over the past three decades, graphene has become the prototypical platform for discovering topological phases of matter. Both the Chern $C \in \mathbb{Z}$ and quantum spin Hall $v \in \mathbb{Z}_2$ insulators were first predicted in graphene, which led to a veritable explosion of research in topological materials. We introduce a new topological classification of two-dimensional matter – the optical *N*-phases $N \in \mathbb{Z}$. This topological quantum number is connected to polarization transport and captured solely by the spatiotemporal dispersion of the susceptibility tensor $\chi$. We verify $N \neq 0$ in graphene with the underlying physical mechanism being repulsive Hall viscosity. An experimental probe, evanescent magneto-optic Kerr effect (e-MOKE) spectroscopy, is proposed to explore the *N*-invariant. We also develop topological circulators by exploiting gapless edge plasmons that are immune to back-scattering and navigate sharp defects with impunity. Our work indicates that graphene with repulsive Hall viscosity is the first candidate material for a topological electromagnetic phase of matter.

[1] School of Electrical and Computer Engineering, Birck Nanotechnology Center, Purdue University, West Lafayette, IN, USA. ✉email: zjacob@purdue.edu

Monolayer graphene forms the canonical system to study two-dimensional (2D) topological phases of matter. The now famous Haldane model of graphene[1], with time-reversal breaking next-nearest-neighbor (NNN) hopping, was the first proposal of a Chern phase $C \in \mathbb{Z}$ and a nontrivial TKNN invariant[2]. Conversely, the Kane-Mele model[3] preserves time-reversal symmetry and was the first example of a quantum spin Hall phase $v \in \mathbb{Z}_2$, resulting from spin–orbit coupling in graphene. Nevertheless, the growing collection of topological phases in condensed matter can be categorized under the umbrella of electrostatics since all observables, e.g., the quantum Hall $\sigma_{xy} = Ce^2/h$ and spin Hall $\sigma_{xy}^s = ve/2\pi$ conductivity, are interpreted at zero photon energy and momentum $\omega = q = 0$. One must go beyond this paradigm to characterize the optical properties of matter, as these are defined for electromagnetic fluctuations over all frequencies $\omega \neq 0$ and momenta $q \neq 0$. We are quickly confronted with two important questions: what are the optical invariants of a material? Do these topological invariants represent unique electromagnetic phases of matter? Our work lays the foundations for this optical classification of condensed matter.

In this paper, we present graphene imbued with Hall viscosity as a paradigmatic example of a topological electromagnetic phase of matter. Please see Fig. 1 and Table 1 comparing the fundamental differences between the Chern phase, quantum spin Hall phase and the optical $N$-phase. This nontrivial topology is revealed in the magnetohydrodynamics of the 2D Navier–Stokes equations. Until quite recently however, viscous electrohydrodynamics with nonzero magnetic field $B \neq 0$ has been experimentally inaccessible, mainly due to impurities and electron–phonon scattering[4]. Exceptional grade 2D materials like graphene[5,6] are providing the first platforms to study the fluidic behavior of electrons[7–9].

Hall viscosity $\eta_H$, the dissipationless component of the viscous stress tensor[10–14], is a generic feature of parity and time-reversal breaking electron fluids and can exhibit quantization analogous to the Hall conductivity[15–17]. It is challenging to measure with conventional techniques but multiple possible observables have been proposed to identify Hall viscosity experimentally, such as negative nonlocal resistance[18,19] and anomalous Hall resistivity[20,21]. Quantization of orbital spin[22] and the Wen-Zee shift[23] represent unique topological numbers of these quantum fluids, which are associated with nontrivial electronic states. Nevertheless, most studies on Hall viscosity have focused on the steady state properties of the material—the optical attributes of

the viscous Hall fluid have remained almost completely unexplored[24–26]. We show that the magnetohydrodynamic response of Hall viscosity gives rise to novel topological phenomena for electromagnetic waves, only manifested in the optical regime.

We show that an optical phase $N \in \mathbb{Z}$ is characterized by the winding number of the susceptibility tensor $\chi$ and intimately related to polarization transport. This topological quantum number is homotopy invariant and found by integrating the material response over all Matsubara frequencies and wavevectors of light. We emphasize that the optical $N$-invariant is distinct from the TKNN invariant, and the first topological quantity that captures the winding in the longitudinal and transverse response functions. Using the $f$-sum rule, we prove that $N$ is generically quantized and immune to perturbations in the optical response. Our definition utilizes a Volovik (Green's function) formalism[27–29], which is naturally generalized to quantum, dissipative and finite temperature systems[30]. Although we only consider the continuum theory here, the formalism is robust and easily extended to the lattice case. We argue that $N$ is the central topological quantity of the electromagnetic linear response theory and classifies all 2D optical media with broken time-reversal symmetry.

We discover that the optical $N$-invariant encodes the vorticity of spin-1 Néel-type skyrmions in the bulk magnetoplasma. This opens the door to experimentally measure the $N$-invariant through a unique magnetic field repulsion reminiscent of the Meissner effect. To probe this deep subwavelength phenomenon, we propose evanescent magneto-optic Kerr effect (e-MOKE) spectroscopy. The angle-resolved Kerr rotation is a direct observable of the skyrmion vorticity and optical $N$-invariant. Lastly, we study the topologically-protected gapless edge states emerging at the boundary of the viscous Hall fluid and vacuum. We demonstrate robust chiral propagation around sharp defects along with back-scatter immunity which we exploit for ultra-subwavelength topological circulators.

Our work unifies the fields of topological photonics and condensed matter physics to spawn a novel area of research in materials science. Although topological photonics[31–34] has mainly focused on artificial media like photonic crystals[35–37] and metamaterials[38], our findings demonstrate that condensed matter can also host topological electromagnetic states. As such, the optical phases we discuss here are microscopic properties of matter and are not related to macroscopic engineering. The optical invariant $N \in \mathbb{Z}$ is therefore a classification of different

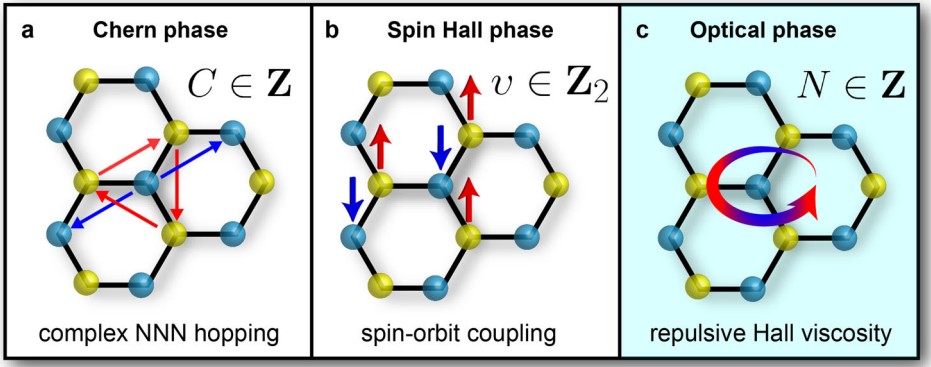

**Fig. 1 Topological phases of graphene. a** The Chern phase $C \in \mathbb{Z}$ arises from complex next-nearest-neighbor (NNN) hopping and is related to charge transport. **b** The quantum spin Hall phase $v \in \mathbb{Z}_2$, also known as the 2D topological insulator, is due to spin–orbit coupling and leads to nontrivial spin transport. **c** The optical phase $N \in \mathbb{Z}$ we put forth in this paper is a consequence of repulsive Hall viscosity and connected to polarization transport. These three phases can be identified as the Chern insulator, quantum spin Hall insulator and viscous Hall insulator, respectively.

**Table 1 Summary of the 2+1D topological phases in graphene.**

| Phase | Chern | Spin Hall | Optical |
|---|---|---|---|
| Quanta | Charge | Spin | Polarization |
| Class | A | All | D |
| Invariant | $C \in \mathbb{Z}$ | $v \in \mathbb{Z}_2$ | $N \in \mathbb{Z}$ |
| Mechanism | Complex NNN hopping | Spin–orbit coupling | Repulsive Hall viscosity |
| Observable | $\sigma_{xy} = Ce^2/h$ | $\sigma_{xy}^s = ve/2\pi$ | Cyclotron null |
| EM field | $\omega = q = 0$ | $\omega = q = 0$ | $\omega \neq q \neq 0$ |

The Chern $C$ and spin Hall $v$ phases are defined at zero photon energy and momentum $\omega = q = 0$. The optical $N$-phases are defined for dynamical electromagnetic (EM) fields $\omega \neq q \neq 0$.

topological phases of matter. In previous photonics systems, Hall viscosity has been absent and therefore are optically trivial $N = 0$. Graphene's viscous Hall fluid is the first candidate for a nontrivial phase $N \neq 0$ and a paradigm shift in optical materials. We note that the optical $N$-invariant captures universal physics beyond graphene and sheds new light on topological superconductors[39], fractional quantum Hall fluids[40], Chern insulators[41], Weyl semimetals[42], and superfluids[43].

## Results

**Dynamics of the viscous Hall fluid.** Our starting point is the 2D Navier–Stokes (NS) equations subject to a uniform magnetic field $B$ and a spatiotemporally varying electric field $\mathbf{E}(t, \mathbf{r})$. The 2D NS equations describe the viscous flow of a parity and time-reversal breaking Hall fluid in Fermi-liquid theory. This theory has successfully explained the experimentally observed steady state properties of graphene[4]. On the other hand, our analysis focuses on the dynamical time-dependent behavior. Viscosity characterizes the resistance to deformation and amounts to a restoring force in the NS equations. The conventional shear viscosity $\eta$ is dissipative and impedes the motion of the fluid. Hall viscosity $\eta_H$ however, is dissipationless and generates a force perpendicular to the motion. Assuming the electric field fluctuations are relatively weak, the charge density $\rho = \rho_0 + \delta\rho$ will be perturbed around its equilibrium value $\rho_0 = -en_0$, where $e$ is the elementary charge and $n_0$ is the electron density. We derive the linearized time-dependent NS equations, which incorporates acceleration $\partial_t \mathbf{J} \neq \mathbf{0}$ and compressibility $\nabla \cdot \mathbf{J} \neq 0$ of the Hall fluid,

$$\partial_t \mathbf{J} = -v_s^2 \nabla \rho - (\gamma - \nu \nabla^2)\mathbf{J} - (\omega_c + \nu_H \nabla^2)\mathbf{J} \times \hat{z} + \frac{e^2 n_0}{m}\mathbf{E}. \quad (1)$$

$\mathbf{E}$ represents the parallel electric field at the location of the electron fluid $z = 0$. The full derivation is provided in Supplementary Note 2. Combining Eq. (1) with the continuity equation $\partial_t \rho + \nabla \cdot \mathbf{J} = 0$ completely specifies the charge $\rho$ and current $\mathbf{J}$ densities with appropriate boundary conditions. The speed of sound is $v_s \simeq v_F/\sqrt{2}$ and we have assumed Dirac dispersion characteristic of graphene[44] to obtain the proportionality of $v_s$ to the Fermi velocity $v_F$. Here, $\gamma = \tau^{-1}$ is the phenomenological damping rate characterizing momentum-non-conserving collisions and $\omega_c = eB/(mc)$ is the cyclotron frequency. $m$ is the effective mass of the electron and $c$ is the speed of light. The kinetic shear and kinetic Hall viscosities are given by $\nu = \eta/(mn_0)$ and $\nu_H = \eta_H/(mn_0)$, respectively. We can also define three important length scales: the shear $D_\nu = \sqrt{\nu\tau}$ and Hall $D_H = \sqrt{|\nu_H/\omega_c|}$ diffusion lengths, as well as the cyclotron radius $r_c = |v_s/\omega_c|$, which characterize the hydrodynamic behavior at mesoscopic scales. The relative sign of $\nu_H$ with respect to $\omega_c$ is paramount to the topological physics and dictates whether Hall viscosity repels or reinforces the magnetic

field. Optical $N$-phases and nontrivial electromagnetic states emerge in the repulsive regime $\omega_c \nu_H > 0$.

**Magnetohydrodynamic susceptibility.** We now derive the bulk linear response theory (LRT) of an unbounded viscous Hall fluid, specifically in the ultra-subwavelength regime. Assuming translational symmetry in the $\mathbf{r} = (x, y)$ plane, the in-plane momentum $\mathbf{q} = (q_x, q_y)$ is conserved which means we can Fourier transform to the reciprocal space. Due to nonlocality arising from pressure $v_s \neq 0$ and viscosity $\nu \neq 0$, the momentum space is particularly useful to understand the LRT. To facilitate this, we utilize the susceptibility tensor $\chi$,

$$\mathbf{P}(\omega, \mathbf{q}) = \chi(\omega, \mathbf{q}) \cdot \mathbf{E}(\omega, \mathbf{q}), \quad (2)$$

which gives the induced polarization density $\mathbf{P}$ to an applied electric field $\mathbf{E}$. The response function $\chi$ completely characterizes the bulk optical properties of the material, for every energy $\omega$ and momentum $\mathbf{q}$ of the photon. Note that both $\omega$ and $\mathbf{q}$ are real parameters here. Exploiting rotational symmetry, we derive the components of the susceptibility tensor in an orthogonal basis,

$$\chi_{ij} = \chi^T(\delta_{ij} - \hat{q}_i \hat{q}_j) + \chi^L \hat{q}_i \hat{q}_j + ig\epsilon_{ij}, \quad (3)$$

where $\hat{q}_i = q_i/q$ is the unit vector directed along the in-plane momentum and $q = \sqrt{\mathbf{q} \cdot \mathbf{q}}$ is its magnitude. $\delta_{ij}$ is the identity and $\epsilon_{ij}$ is the 2D Levi-Civita symbol. In traditional LRT ($q \approx 0$), this crucial distinction between longitudinal and transverse response functions is ignored. However, the optical $N$-invariant captures the deep subwavelength topology of matter and requires both the current–current (transverse) and density–density (longitudinal) response. The transverse $\chi^T$ and longitudinal $\chi^L$ response functions can be identified in the induced polarization using a transverse $\mathbf{q} \cdot \mathbf{E} = 0$ and longitudinal $\mathbf{q} \times \mathbf{E} = 0$ electric field respectively. Gyrotropy $g$ couples these two components.

Note, the response function is temporally dispersive ($\omega$ dependent) as well as spatially dispersive ($\mathbf{q}$ dependent), and both properties are essential to realize optical $N$-phases. Temporal dispersion quantifies the degrees of freedom of the electronic excitations, while spatial dispersion characterizes the geometric phase of the induced polarization. We decompose the components of $\chi$ into its transverse $\chi^T$,

$$\chi^T = -\frac{e^2 n_0}{m} \frac{1}{\omega\tilde{\omega}} \left(1 + \frac{\omega\Omega_c^2}{\tilde{\omega}(\omega\tilde{\omega} - v_s^2 q^2) - \omega\Omega_c^2}\right), \quad (4a)$$

longitudinal $\chi^L$,

$$\chi^L = -\frac{e^2 n_0}{m} \frac{\tilde{\omega}}{\tilde{\omega}(\omega\tilde{\omega} - v_s^2 q^2) - \omega\Omega_c^2}, \quad (4b)$$

and gyrotropic $g$ response,

$$g = \frac{e^2 n_0}{m} \frac{\Omega_c}{\tilde{\omega}(\omega\tilde{\omega} - v_s^2 q^2) - \omega\Omega_c^2}. \quad (4c)$$

$\tilde{\omega} = \omega + i\Gamma$ is the shifted energy, where $\Gamma(q) = \gamma + \nu q^2$ is the viscous damping rate that describes the decay pathways. $\Omega_c(q)$ is the viscous cyclotron frequency,

$$\Omega_c(q) = \omega_c - \nu_H q^2. \quad (5)$$

Due to Hall viscosity $\nu_H$, the effective magnetic field in the Hall fluid is momentum dependent $B_{eff}(q) = mc\Omega_c(q)/e = B(1 - D_H^2 q^2)$ and varies on the scale of the Hall diffusion length $D_H$. In the dissipationless (Hermitian) limit $\Gamma \to 0$, we obtain the LRT of an ideal quantum Hall fluid $\chi = \chi^\dagger$.

Lastly, we verify that the susceptibility tensor satisfies the reality condition,

$$\chi(\omega, \mathbf{q}) = \chi^*(-\omega, -\mathbf{q}), \quad (6)$$

since electromagnetism is a real-valued vector field theory. Due to Eq. (6), the components of $\chi$ cannot be completely independent, implying the excitations belong to universality class D, the same symmetry class as topological superconductors[45]. This should be contrasted with the electron, which belongs to symmetry class A, or class AII in the presence of time-reversal symmetry. Particle number is conserved in the complex fermionic classes due to U(1) symmetry, but this is not true for the bosonic classes since the excitations are their own antiparticle. Generically, the spatiotemporally dispersive susceptibility tensor $\chi(\omega, \mathbf{q})$ represents a mapping from the 2 + 1D momentum space to the general real linear group $\text{GL}_n(\mathbb{R})$. $n$ denotes the degrees of freedom—i.e., the total $n \times n$ matrix dimension of $\chi$.

**Optical $N$-invariant**. The susceptibility tensor $\chi$ is precisely the Green's function of the polarization density $\mathbf{P}$ and is therefore a topological object. The cornerstone of the Green's function approach developed by Volovik[27–29], lies the following 2 + 1D topological invariant,

$$N = \frac{\epsilon^{\alpha\beta\gamma}}{24\pi^2} \int d\Omega d\mathbf{q} \, \text{tr} \left[ \chi \frac{\partial \chi^{-1}}{\partial q_\alpha} \chi \frac{\partial \chi^{-1}}{\partial q_\beta} \chi \frac{\partial \chi^{-1}}{\partial q_\gamma} \right], \quad (7)$$

where $\chi(\omega, \mathbf{q}) \to \chi(\Omega, \mathbf{q})$ is parameterized by the complex frequency variable $\omega \to \Omega$. In this case, $\partial_\alpha$ indicates partial derivatives with respect to the total momentum coordinate $q_\alpha = (\Omega, \mathbf{q})$ and tr denotes the trace over the tensor indices of $\chi$. Note that the temporal integral $d\Omega$ is performed vertically over all imaginary (Matsubara) frequencies[46],

$$\Omega \in (\omega - i\infty, \omega + i\infty). \quad (8)$$

$\omega = \Re(\Omega)$ is the photon energy that is assumed to lie within the electronic band gap $0 < \hbar\omega < E_{\text{bg}}$. Bulk current cannot be generated since the photon does not possess sufficient energy to stimulate a transition – it can only polarize the material. With parabolic dispersion ($2\omega_c\nu_H < v_s^2$), the band gap of an ideal quantum Hall fluid is defined by the first Landau level ($E_{\text{bg}} = \hbar|\Omega_c(0)| = \hbar|\omega_c|$). A plot of the contour integral is shown in Fig. 2b.

Equations (7) and (8) clearly highlight the differences from the Chern number (TKNN invariant) and previous theories in the field of topological photonics. We reiterate that the optical $N$-invariant is fundamentally different than the electronic Chern number as they describe physically distinct quantities. The Chern

number $C$ is related to the U(1) Berry phase of the wave function $\psi$. On the other hand, the $N$-invariant quantifies the spectral asymmetry in the susceptibility tensor $\chi$. Accordingly, the Chern number $C$ characterizes the topology of the bulk electronic band structure while the $N$-invariant characterizes the topology of the bulk polaritonic band structure[25,26]. These are plasmons for the quantum Hall fluid, but all dipole-carrying excitations such as excitons, phonons, magnons, and Cooper-pairs[47] can be accounted for in our theory.

To prove $N$ is a quantized topological invariant, we consult a few fundamental properties of the response function. Every solid is transparent at $|\Omega| \to \infty$ since all electrons respond as free particles to rapidly varying temporal oscillations. The susceptibility tensor approaches a purely diamagnetic response which is independent of $\mathbf{q}$ and nonsingular $\det \chi \neq 0$,

$$\lim_{|\Omega| \to \infty} \chi_{ij}(\Omega, \mathbf{q}) \to -\frac{e^2 n_0}{m\Omega^2} \delta_{ij}. \quad (9)$$

This is a universal property of optical materials and ultimately a consequence of the $f$-sum rule, also known as the optical, conductivity, or Thomas-Reiche-Kuhn (TRK) sum rule[48]. The $f$-sum rule guarantees that our contours in the complex frequency plane are nondegenerate which is essential for homotopy invariance[29,46]. The detailed proof is presented in Supplementary Note 3. By including the point at infinity ($|\Omega| = \infty$), each contour in Eq. (8) defines a circle $S^1$ in the extended complex plane. Moreover, due to viscosity, the susceptibility tensor is naturally regularized and approaches a directionally independent value as $q \to \infty$,

$$\lim_{q \to \infty} \chi(\Omega, \mathbf{q}) \to \chi(\Omega, q). \quad (10)$$

Again, by including the point at infinity ($q = \infty$), the $\mathbf{q}$ space is topologically equivalent to the sphere $S^2$. The combined 2 + 1D momentum space is compactified and effectively $S^2 \times S^1$. A visualization of this manifold is depicted in Fig. 2a. The susceptibility tensor $\chi(\Omega, \mathbf{q})$ is therefore an element of the third homotopy group of $\text{GL}_n(\mathbb{R})$,

$$\pi_3[\text{GL}_n(\mathbb{R})] = \mathbb{Z}, \quad (11)$$

which is isomorphic to $\mathbb{Z}$. Equation (11) epitomizes the crucial fact that the optical $N$-invariant is universal and model-independent. The integral in Eq. (7) calculates the precise element of $N \in \mathbb{Z}$ the response function corresponds to, where

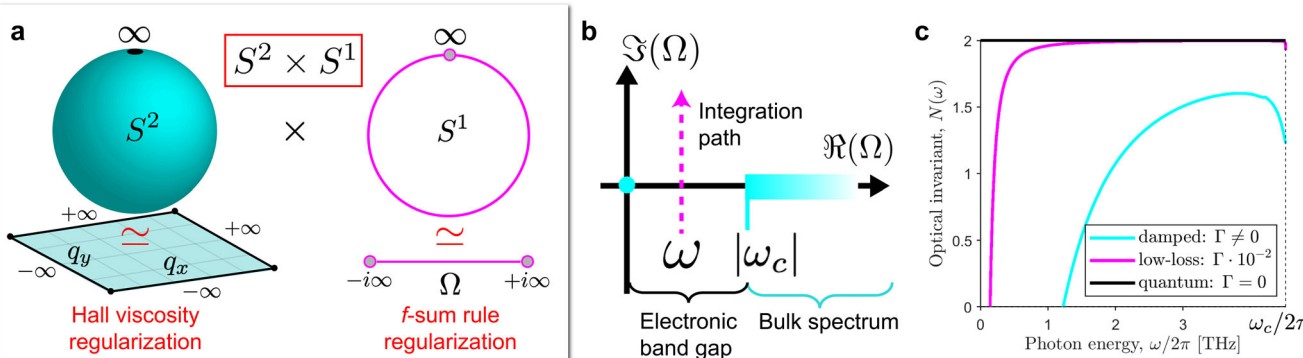

**Fig. 2 Visualization of the 2 + 1D momentum space in the continuum limit. a** Due to regularization from viscosity, the $\mathbf{q}$ space is equivalent to the sphere $\mathbb{R}^2 \simeq S^2$ as all paths at $q = \infty$ are compactified into a single point. Likewise, the $f$-sum rule ensures imaginary contours in $\Omega$ space are compactified to a circle $\mathbb{R} \simeq S^1$. The product of these two spaces $S^2 \times S^1$ is a 2 + 1D manifold without boundary. **b** Contour in the complex frequency plane used to evaluate the optical $N$-invariant. Integration is performed vertically $\Omega \in (\omega - i\infty, \omega + i\infty)$ over all Matsubara frequencies. $\omega = \Re(\Omega)$ is the photon energy that lies within the electronic band gap $0 < \omega < |\omega_c|$. **c** Optical invariant $N(\omega)$ as a function of the photon energy $\omega$. The $N$-invariant is calculated for damped, low-loss, and quantum Hall fluids in the nontrivial repulsive regime $\omega_c\nu_H > 0$. In the dissipationless limit $\Gamma = 0$, the optical invariant is quantized to $|N| = 2$ within the entire band gap.

each integer represents a unique optical phase. This should be contrasted with the generalized TKNN invariant $C$, which is characterized by complex fields $\pi_3[\mathrm{GL}_n(\mathbb{C})] = \mathbb{Z}$. We stress that although both $N$ and $C$ are integer invariants, the symmetry classes are fundamentally different.

**Optical $N$-invariant of the viscous Hall fluid**. We verify that graphene's viscous Hall fluid is the first candidate material for a topological electromagnetic phase of matter. Specifically, it has nontrivial optical invariant ($N \neq 0$) when Hall viscosity is repulsive $\omega_c \nu_H > 0$. Utilizing the magnetohydrodynamic theory [Eq. (4)] and the homotopy equation [Eq. (7)] we arrive at,

$$N = \int \frac{d\Omega d\mathbf{q}}{i 2\pi^2} \frac{v_s^4 q^2}{\Omega} \frac{[\Omega(\omega_c + \nu_H q^2) + i(\gamma\nu_H + \omega_c \nu)q^2]}{[\tilde{\Omega}(\Omega\tilde{\Omega} - v_s^2 q^2) - \Omega\Omega_c^2]^2}, \quad (12)$$

where $\tilde{\Omega} = \Omega + i\Gamma$ and the sign of the dissipation is implied $\Gamma \to \mathrm{sgn}[\Im(\Omega)]\Gamma$ to preserve causality. We integrate Eq. (12) numerically in Fig. 2c but it is important to confirm that $N$ is quantized in the zero temperature quantum limit $\Gamma \to 0$. Integrating over all Matsubara frequencies and wavevectors, we acquire the optical $N$-invariant of the viscous Hall fluid,

$$N = \mathrm{sgn}(\omega_c) + \mathrm{sgn}(\nu_H). \quad (13)$$

The invariant is quantized to $N = \pm 2$ when Hall viscosity is repulsive $\omega_c \nu_H > 0$ and trivial $N = 0$ for $\omega_c \nu_H < 0$. We note that the definition of $N$ through the homotopy group [Eq. (7)] is robust and does not rely on any symmetries of the NS theory such as Galilean invariance and frequency-momentum independence in the viscosity coefficients. Thus, the $N$-invariant can be immediately generalized beyond graphene to more exotic materials; for instance the magneto-roton excitations in fractional quantum Hall fluids[24,49].

**e-MOKE spectroscopy and skyrmion texture**. Here we put forth an experimental probe of the optical $N$-invariant. The $N$-invariant encodes a peculiar vorticity in the bulk magnetoplasmons (BMPs) that arises from repulsive Hall viscosity ($\omega_c \nu_H > 0$). We prove these are spin-1 skyrmions[50] carrying deep subwavelength texture of the optical $N$-invariant. The fundamental reason why they are spin-1 is because electromagnetism is a vector field theory so spin arises from the circular polarization state of $\mathbf{P}$. The self-consistent solutions of Poisson's equation and the NS equations [Eq. (1)] produces the BMP dispersion $\omega = \omega_b(q)$. A short review is provided in the Supplementary Note 5. At low $q$, we obtain the characteristic square root dispersion ($\sqrt{q}$) of a BMP which is generated from the long-range Coulomb interaction. However, the hydrodynamic behavior dominates at high $q$ and is the origin of topologically nontrivial electromagnetic states.

Although BMPs in quantum Hall fluids have been studied for many years, the effect Hall viscosity has on dynamical variables like the angular momentum (AM) has never been considered. From first principles electrodynamics, we derive the energy density $u_\mathbf{q}$ and AM density $m_\mathbf{q}$ in the bulk momentum space. The ratio of these two quantities gives the AM $j_z$ per unit energy $\omega_b$ of the BMP,

$$\frac{m_\mathbf{q}}{u_\mathbf{q}} = -\left[\frac{\partial g}{\partial \omega} \Big/ \frac{\partial(\omega\chi^L)}{\partial \omega}\right]_{\omega=\omega_b} = \frac{j_z}{\omega_b}. \quad (14a)$$

$j_z$ is the AM of a spin-1 Néel-type skyrmion[51] which varies dynamically with the momentum $q$,

$$j_z = \frac{\Omega_c}{\omega_b}\left(\frac{2\omega_b^2 + \Omega_P^2}{2\omega_b^2 - \Omega_P^2}\right). \quad (14b)$$

$\Omega_P$ being the effective plasma frequency. As a consequence of

rotational symmetry, the skyrmion number is determined by the difference in AM eigenvalues $\Delta j_z$ at the rotationally invariant momentum $q = 0$ and $q = \infty$. The eigenvalues must be integers $j_z = \pm 1$ or $0$ at these points due to the spin-1 representation[25]. For the BMP we obtain,

$$\Delta j_z = j_z(0) - j_z(\infty) = \mathrm{sgn}(\omega_c) + \mathrm{sgn}(\nu_H). \quad (15)$$

Notice that the AM flips direction giving $|\Delta j_z| = 2$ in the repulsive regime $\omega_c \nu_H > 0$—the signature of a topologically nontrivial skyrmion. Indeed, the skyrmion winding number is exactly equal to the optical $N$-invariant derived in Eq. (13),

$$N = \Delta j_z. \quad (16)$$

It should be noted that Eq. (16) only holds in the presence of rotational symmetry, nonetheless the quantization of $N$ remains robust even without symmetry [Eq. (11)]. In this case, the $N$-invariant is solely determined by the $j_z$ eigenvalues at high-symmetry points $q = 0$ and $q = \infty$ which does not rely on the specific physical model of Hall viscosity. The dynamical skyrmion texture encodes a "knot" in the polarization density that cannot be undone through any continuous deformation. The AM texture $j_z(q)$ as a function of the in-plane momentum $q$ is depicted in Fig. 3e.

As can be seen directly from Fig. 3e, the optical $N$-invariant and skyrmion number is only nontrivial ($N = \Delta j_z \neq 0$) when the viscous cyclotron frequency changes sign [Eq. (5)]. We define this phenomenon as the cyclotron null ($\Omega_c(q) = 0$) which occurs at a particular in-plane momentum $q = D_H^{-1}$ and cannot be removed unless there is a topological phase transition. $q = D_H^{-1}$ is the momentum where the effective magnetic field is completely expelled from the fluid ($B_{\mathrm{eff}}(q) = 0$) and is unique to the nontrivial repulsive regime $\omega_c \nu_H > 0$. At the null, the cyclotron motion switches handedness and the circulating currents appear to rotate in the opposite direction. This interesting phenomenon is reminiscent of the Meissner effect in a superconductor, that causes all magnetic fields to be expelled from the electron fluid. The difference is that the cyclotron null $\Omega_c(q) = 0$ is a deep subwavelength effect as it is a consequence of Hall viscosity.

Our goal is to measure the optical $N$-invariant in bulk graphene through this unique magnetic field repulsion. These topological properties are revealed in the Kerr rotation of reflected photons. The Faraday and Kerr effects have been utilized in condensed matter physics to measure the quantized magnetoelectric response in 3D topological insulators like $Bi_2Se_3$[52]. However, all such experiments have been performed with low momentum electromagnetic waves at small incident angles ($q \approx 0$), which cannot probe Hall viscosity or the $N$-invariant. In fact, evanescent waves are necessary $q \gg 2\pi/\lambda$ since the Hall diffusion length $D_H \ll \lambda/2\pi$ is generally much smaller than the wavelength of light $\lambda$.

To overcome this obstacle, we propose evanescent magneto-optic Kerr effect (e-MOKE) spectroscopy—an angle-resolved and high-momentum probe of the optical $N$-invariant. A schematic of the e-MOKE system is displayed in Fig. 3a. The polar configuration is graphene on substrate with an applied $B$-field and the top exposed to vacuum. For incident light, we exploit a high index prism available at THz frequencies to interface total internally reflected evanescent waves with the viscous Hall fluid. Although the local interaction at the sample location has large momentum, the subwavelength gyrotropic information is carried to the far field by reflected photons facilitating the read-out of e-MOKE data through traditional lock-in techniques. The experimental smoking gun of $B$-field repulsion and a nontrivial $N$-invariant is the switch in ellipticity of reflected light, depicted in Fig. 3b. This vanishing Kerr rotation can be measured even at

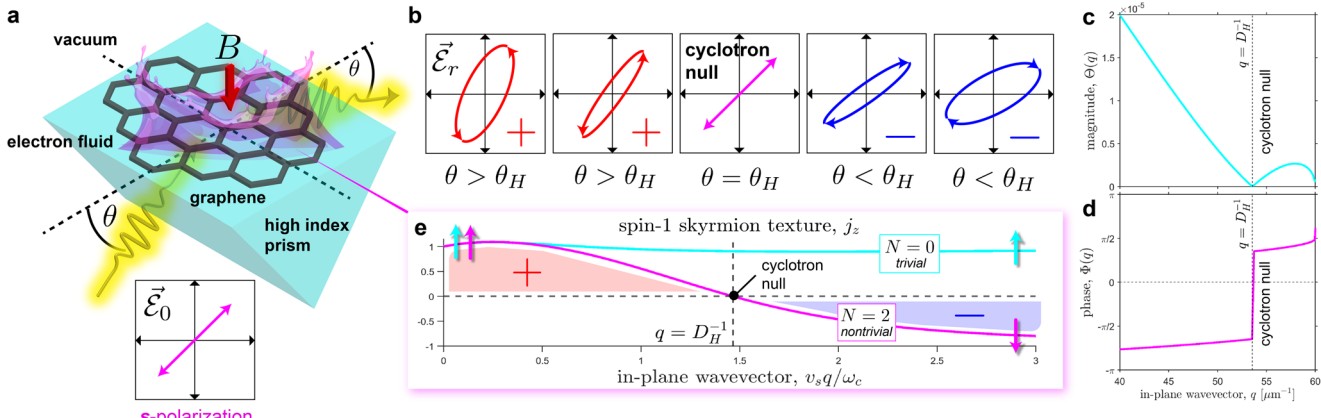

**Fig. 3 Experimental e-MOKE setup of topological viscous Hall fluid. a** Overview of e-MOKE spectroscopy for direct measurement of the optical $N$-invariant through $B$-field repulsion. **b** Evolution of the reflected polarization ellipse $\vec{\mathcal{E}}_r$ due to an incident $\hat{s}$ polarized wave $\vec{\mathcal{E}}_0$ at the e-MOKE interface. The frequency of incident light is $\omega/2\pi = 286$ THz. The ellipticity gradually changes for various incident angles ($\theta$) but abruptly switches handedness due to skyrmionic vorticity. This cyclotron null coincides with an incident angle $\tan\theta_H = \sqrt{(2\pi n_- D_H/\lambda)^2 - 1}$ and only occurs in the nontrivial phase $N = 2$. **c, d** Magnitude $\Theta$ and phase $\Phi$ of the Kerr rotation plotted against the in-plane momentum $q$. **e** The optical $N$-invariant encodes the angular momentum texture of spin-1 Néel-type skyrmions and is identified with the skyrmion winding number $\Delta j_z = N$. This is an experimentally measurable signature of the $N$-invariant. In the nontrivial phase $N = 2$, the spin flips direction which is indicated by the arrows at the high-symmetry points $q = 0$ and $q = \infty$.

room temperature. Theoretical plots of the Kerr rotation are displayed in Fig. 3c, d.

**Topological edge magnetoplasmons and circulators.** We now uncover the topological edge states emerging at the boundary of the viscous Hall fluid ($N = 2$) and show they are incredibly robust, impervious to boundary effects, as well as immune to back-scattering. We strongly emphasize that the edge states considered here are fundamentally different than traditional edge magnetoplasmons (EMPs) predicted by Fetter[53], because they are gapless and only manifest in the magnetohydrodynamic regime. Topological EMPs with Hall viscosity cannot be regarded as a correction to gapped Fetter EMPs, since they cannot be continuously deformed into one another. This is demonstrated explicitly in Supplementary Note 6.

The extraordinary property of topological EMPs is that they are completely unaffected by boundary slip. The electron fluid exerts zero shear stress on the boundary (stress-free) while simultaneously having zero tangential current (no-slip). This is very peculiar because slip usually plays a significant role in the boundary layer physics. Indeed, many factors can alter the magnitude of slip such as surface roughness, interface-fluid attraction, nanobubble nucleation and high shear rates[54], but the dispersion of the topological EMP remains independent. Note that this phenomenon is only possible in the nontrivial phase $|N| = 2$ and absolutely no edge state exists in the trivial regime $N = 0$. Visualizations of the hydrodynamic boundary conditions are shown in Fig. 4a with a comparison to traditional magneto-optics in Fig. 4b. The dispersion relations of the BMP $\omega_b(q_y)$ and topological EMP $\omega_e(q_y)$ are displayed in Fig. 4c. We have calculated the dispersion for arbitrary boundary slip conditions to demonstrate the robustness of the edge state.

Lastly, we numerically simulate an ultra-subwavelength topological circulator and illustrate the robust unidirectional transport of topological edge plasmons in complicated geometries. We choose a hexagonal sample of viscous Hall fluid with a center to edge length of $d = 20r_c$, where $r_c = |v_s/\omega_c|$ is the cyclotron radius. To excite the topological EMPs we let the dipole source oscillate in the band gap $0 < \omega < |\omega_c|$ such that no bulk current can be generated. The normalized charge density

fluctuation $\delta\rho/\rho_0$ is plotted in Fig. 4d. For an excitation frequency of $\omega = |\omega_c|/2$, the edge wave propagates at approximately the speed of sound $v_s$. Interestingly, the edge state propagates around sharp defects with zero back-scattering and is impervious to boundary effects. Supplementary Movie 1 showcases the propagation of the edge state around a sharp defect. Supplementary Fig. 2 displays the geometry of a 3-port topological circulator.

## Discussion

We have introduced the optical phases $N \in \mathbb{Z}$ of two-dimensional quantum matter—a topological classification emerging from the invariant optical proprieties of a material. As a particularly important platform, we have shown that $N \neq 0$ is nontrivial in graphene's viscous Hall fluid and fundamentally tied to repulsive Hall viscosity. In addition, we have proposed a unique probe of topological matter: evanescent magneto-optic Kerr effect (e-MOKE) spectroscopy to search for nontrivial spin-1 skyrmion texture and the $B$-field expulsion that is reminiscent of the Meissner effect. Robustness and back-scatter immunity of the topological edge states was demonstrated by analyzing an ultra-subwavelength circulator. These intriguing optical $N$-phases are also expected in topological superconductors, fractional quantum Hall fluids, Chern insulators, Weyl semimetals and superfluids[39–43], leading to a new generation of effects at the interface of topological photonics and condensed matter physics.

## Methods

**Energy regularization from the *f*-sum rule.** The *f*-sum rule [Eq. (9)] is a form of energy conservation on light-matter interactions. It implies the sum over all electronic transitions must equal the total energy put into the system. This places a fundamental constraint on the electromagnetic LRT. Over the entire frequency space, the susceptibility tensor can be expressed as $\chi = \chi' + i\chi''$ with $\chi'$ and $\chi''$ both Hermitian. $\chi''$ accounts for dissipation and absorption mechanisms from electron transitions. In a quantum system, $\chi''$ is zero in the band gap $0 < \hbar\omega < E_{bg}$ but contains $\delta$-functions at the transition energies. In any case, the absorption over all $\omega$ is invariant due to the *f*-sum rule[48],

$$\int_{-\infty}^{\infty} \omega \chi''_{ij}(\omega, \mathbf{q})d\omega = \int_{-\infty}^{\infty} \sigma'_{ij}(\omega, \mathbf{q})d\omega = \pi\frac{e^2 n_0}{m}\delta_{ij}, \quad (17a)$$

where $\sigma'$ is the dissipative part of the conductivity tensor $\sigma = \sigma' + i\sigma''$. It is clear that Eq. (17a) can never vanish, otherwise transitions would cost no energy. Using the Kramers–Kronig relations for large $|\Omega| \to \infty$, we immediately obtain the

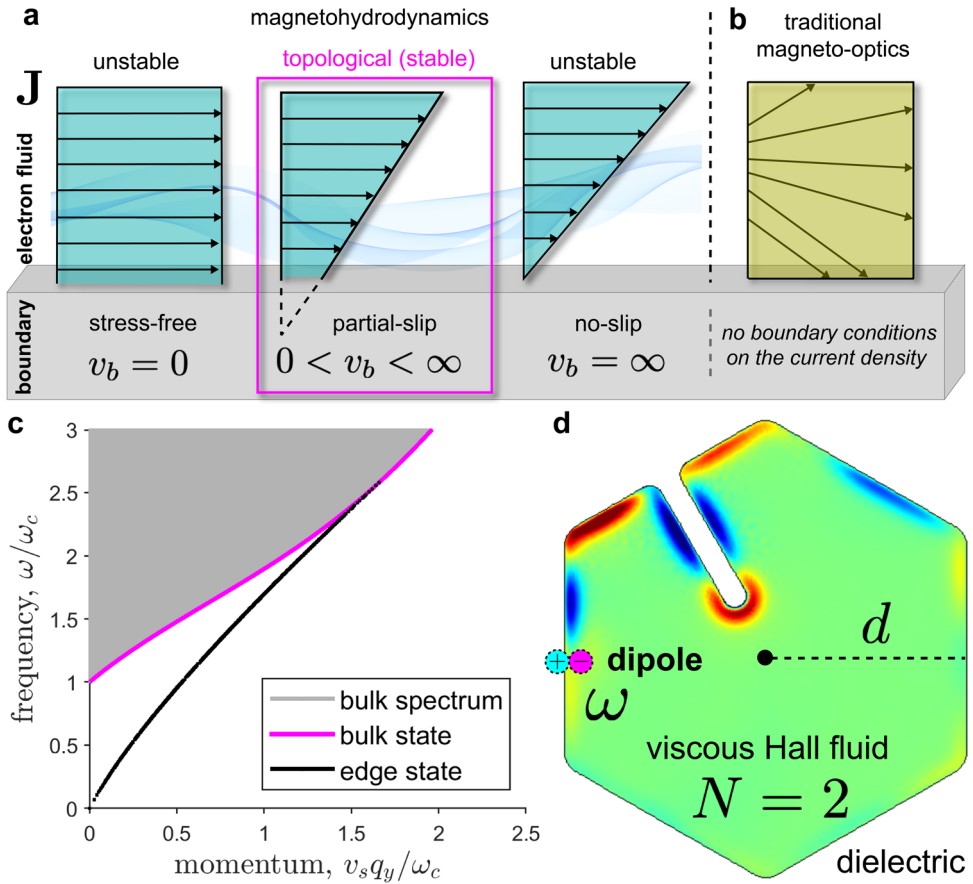

**Fig. 4 Boundary layer of the topological viscous Hall fluid. a** Depiction of the additional boundary conditions (ABCs) on the current density **J** in magnetohydrodynamic systems. The boundary scattering velocity $v_b$ dictates the amount of slip at the interface, with $v_b = 0$ and $v_b = \infty$ being the extreme cases of stress-free and no-slip BCs respectively. **b** Traditional magneto-optics does not consider nonlocal phenomenon and the associated ABCs on **J**. This theory cannot describe the physics in the hydrodynamic boundary layer. **c** Bulk and edge dispersion of the dissipationless $\Gamma = 0$ nontrivial $N = 2$ viscous Hall fluid. No edge states exist in the trivial regime $N = 0$. Magenta and black lines are the BMP and topological EMP dispersion respectively, $\omega_b(q_y)$ and $\omega_e(q_y)$. Gray denotes the continuous bulk spectrum. The simulations include the Coulomb interaction and we have imposed arbitrary boundary slip conditions. Topological EMPs exist for all values of the scattering velocity $v_b$. **d** Numerical simulation of the topological circulator on dielectric substrate. The center to edge length is $d = 20r_c = 548$ nm. A dipole source is placed at the boundary and oscillates periodically in the band gap at $\omega/2\pi = |\omega_c|/4\pi = 2.26$ THz for a total of 1.7 ps. The color plot shows the normalized charge density fluctuation $\delta\rho/\rho_0$ where red and blue indicate positive and negative values respectively. The topological edge state navigates sharp defects with zero back-scattering and is immune to boundary effects.

asymptotic behavior of $\chi$,

$$\lim_{|\Omega| \to \infty} \chi_{ij}(\Omega, \mathbf{q}) \to -\frac{1}{\pi\Omega^2} \int_{-\infty}^{\infty} \sigma'_{ij}(\omega, \mathbf{q}) d\omega = -\frac{e^2 n_0}{m\Omega^2} \delta_{ij}. \tag{17b}$$

Topologically, the $f$-sum rule can be understood as an energy ($\Omega$) regularization. Since the susceptibility asymptotically approaches the identity, all singularities (poles and zeros) of $\chi$ occur at finite values of $\Omega$. Hence, any sufficiently large contour in the complex plane is nondegenerate which is a necessary condition for homotopy invariance.

**Topological stability of the optical N-phases.** Here we demonstrate explicitly that $N$ is a quantized topological invariant and stable under variations. To prove this, we need only consult the $f$-sum rule and the regularity of $\chi$ at $q \to \infty$, outlined in Eq. (9) and (10) respectively. This definition generalizes beyond the NS equations to encompass all continuous optical media. We consider an arbitrary perturbation to the response function $\chi \to \chi + \delta\chi$, which takes $N \to N + \delta N$,

$$\delta N = -\frac{\epsilon^{\alpha\beta\gamma}}{8\pi^2} \int d\Omega d\mathbf{q} \, \partial_\alpha \text{tr} \left[ \delta\chi \partial_\beta \chi^{-1} \chi \partial_\gamma \chi^{-1} \right]. \tag{18}$$

All perturbations amount to a total divergence in the integrand. Since the susceptibility tensor is regular at $q \to \infty$, the spatial **q** boundary terms necessarily vanish. This is equivalent to the periodic boundary condition in the lattice theory. However, temporal $\Omega$ boundary terms can contribute if the system is non-Hermitian (dissipative) since $\chi$ is discontinuous along the imaginary line of $\Omega$. This is observed in Fig. 2c. In the dissipationless quantum limit $\Gamma \to 0$, the response function is Hermitian $\chi = \chi^\dagger$ and therefore continuous along the entire imaginary

line. Due to the $f$-sum rule, the temporal boundaries are zero and $\delta N = 0$ vanishes identically. $N \in \mathbb{Z}$ is thus topologically quantized and immune to variations in the optical response.

**e-MOKE setup and read out.** The MOKE spectroscopy setup we consider is in polar geometry, with the external magnetic field $B$ applied normal to the graphene sample. Since graphene is a 2D material, this experiment is more specifically known as the surface MOKE (SMOKE)[55]. The Kerr effect, or Kerr rotation, is the phenomenon where linearly polarized light becomes elliptical upon reflection. The cross polarization term $r_{ps}$ that couples $\hat{s}$ and $\hat{p}$ polarized waves is directly proportional to the gyrotropy $r_{ps} \propto g \propto \Omega_c$ and a definite measurement of the Kerr rotation. We incident THz frequency light on the sample at steep angles to probe the high-momentum modes of the electron fluid. To determine if the sample is topologically nontrivial, we must look for nulls in the Kerr rotation,

$$\tan\left(\frac{\Theta}{2}\right) \exp(i\Phi) = \frac{r_{ps}}{r_{ss}}. \tag{19}$$

The exact expressions for the reflection coefficients are derived in the Supplementary Note 4 along with a short review of boundary conditions on 2D charge densities in Supplementary Note 1. The Kerr rotation is defined by the ratio of reflected $\hat{p}$ to $\hat{s}$ polarization due to an incident $\hat{s}$ polarized wave. $\Theta \in [0, \pi]$ is the relative magnitude and $\Phi \in [-\pi, \pi]$ is the relative phase. Theoretical plots of the MOKE are displayed in Fig. 3c, d. We explicitly calculate the relevant parameters for monolayer graphene under conventional laboratory settings to isolate the frequency-momentum space that should be explored for topological phenomena [see Table 2].

**Table 2 Parameters for monolayer graphene at room temperature.**

| | |
|---|---|
| Lattice constant, $a$ | 2.46 Å |
| Electron density, $n_0$ | $2 \times 10^{12}$ cm$^{-2}$ |
| Effective electron mass, $m$ | $0.0124 m_e$ |
| Fermi velocity, $v_F$ | $1.1 \times 10^6$ m/s |
| Kinematic shear viscosity (zero bias), $\nu_0$ | 0.1 m²/s |
| Intrinsic magnetic field, $B_0$ | 0.2 T |
| Biasing magnetic field, $B$ | 2 T |
| Transport time, $\tau$ | 2 ps |
| Kinetic shear viscosity, $\nu$ | $9.90 \times 10^{-4}$ m²/s |
| Shear diffusion length, $D_\nu = \sqrt{\nu \tau}$ | 44.5 nm |
| Cyclotron frequency, $\omega_c/2\pi$ | 4.52 THz |
| Kinetic Hall viscosity, $\nu_H$ | $9.90 \times 10^{-3}$ m²/s |
| Hall diffusion length, $D_H = \sqrt{|\nu_H/\omega_c|}$ | 18.7 nm |
| Cyclotron radius, $r_c = |v_s/\omega_c|$ | 27.4 nm |
| Vacuum index, $n_+$ | 1 |
| Substrate index, $n_-$ | 10 |

**Boundary conditions on viscous Hall fluids.** An important feature of an electron fluid, as opposed to an ordinary fluid, is that it interacts through the Coulomb potential. Accordingly, we must ensure continuity of the electric potential $\phi$ and its normal derivative $\hat{n} \cdot \nabla \phi$ at the boundary. Although edge states in chiral active fluids have been studied in the context of the NS equations[56,57], scarce few works have coupled them to Poisson's equation[58]. The most significant impact of the long-range Coulomb interaction is that at low momentum $q_y \to 0$, the group velocity of the EMP diverges logarithmically $\partial \omega_e / \partial q_y \sim -\log q_y$, which is characteristic of a 1D plasmon[59].

Due to nonlocality in a hydrodynamic electron fluid[60], we also require boundary conditions on the current density **J**. In optics these are known as additional boundary conditions (ABCs)[61] since the charge configuration at any point is not uniquely determined by the electric field at the same point. Nonlocal theories were first utilized to understand the anomalous skin effect in metals at high frequencies[62]. We show that a topological skin effect occurs in the nontrivial phase $|N| = 2$, where an AC current propagates unidirectionally at the edge and decays exponentially into the bulk. The BCs on **J** follow from elementary conservation laws. Charge conservation necessitates a vanishing normal current,

$$\hat{n} \cdot \mathbf{J}|_{\partial V} = 0, \tag{20a}$$

where $\hat{n}$ is the outward normal unit vector and $\partial V$ denotes the boundary of the volume $V$. A viscous hydrodynamic fluid also requires a BC on the tangential current, which is related to momentum conservation. The tangential current is proportional to the shear stress (off-diagonal stress) on $\partial V$[21],

$$\left[ -e\hat{t} \cdot \varsigma \cdot \hat{n} + m v_b \hat{t} \cdot \mathbf{J} \right]_{\partial V} = 0. \tag{20b}$$

$\hat{t} = \hat{n} \times \hat{z}$ is the unit tangential vector, $\varsigma$ is the viscous stress tensor and $v_b$ is the boundary scattering velocity that dictates the slip flow. Note that both shear $\nu$ and Hall $\nu_H$ viscosity generate a shear stress on the boundary so a frictional force remains even in the quantum limit $\nu \to 0$.

The scattering velocity $v_b \geq 0$ is generally positive for electron fluids but can be negative in principle[63]. In the extreme limits of $v_b = 0$ and $v_b = \infty$, we recover the familiar stress-free and no-slip BCs respectively. Although no-slip is approximately valid at macroscopic scales, it has no microscopic justification and is normally not satisfied. We must consider the partial-slip scenario $0 < v_b < \infty$ to accurately describe the boundary physics. A visualization of the various hydrodynamic BCs is depicted in Fig. 4a. Another significant issue with stress-free and no-slip is that they can generate unstable spurious solutions[64,65]. These unphysical states will be gapped out immediately if we continuously deform the BC. The topological ones, however, will remain. This is the case for topological EMPs which is shown in Supplementary Fig. 1. In the nontrivial phase $|N| = 2$, the EMP satisfies no-slip and stress-free BCs simultaneously and is therefore independent of the boundary slip flow. Indeed, Eq. (20b) is fulfilled automatically for any value of $v_b$, including negative values. This interesting property is demonstrated in Fig. 4c, d, which is simulated for arbitrary values of $v_b$.

## Data availability
The authors declare that the data supporting the findings of this study are available within the paper and its supplementary information files.

## Code availability
The code compiled for numerical simulations are available from the corresponding author upon reasonable request.

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

## Acknowledgements

This research was supported by the Defense Advanced Research Projects Agency (DARPA) Nascent Light-Matter Interactions (NLM) Program.

## Author contributions

T.V.M. conceived of the idea and developed the theoretical treatment. W.S. designed the circulator and performed the numerical simulations. Z.J. proposed the experiment, helped interpret the mathematics and supervised the project. All authors contributed to discussions, results and development of the manuscript.

## Competing interests

The authors declare no competing interests.
