## [Peer Review File · Nature Communications]

REVIEWERS' COMMENTS

Reviewer #1 (Remarks to the Author):

The authors have convincingly addressed my suggestions and questions. The manuscript has considerably improved and in my opinion it is ready for publication.

Reviewer #2 (Remarks to the Author):

In their revised version, the authors have added several new details and discussions to help address my earlier concerns. In particular, the proof of nondegeneracy of the susceptibility tensor using the f-sum rule, and the relationship of the N-invariant to the magnetoplasmon angular momentum texture are particularly convincing and illuminating. I am pleased to say that I am now willing to recommend this manuscript for publication in Nature Physics.

I have one remaining concern, related to the use of rotational symmetry in the derivation of the skyrmion angular momentum. Although the quantization of the N-invariant does not require rotational symmetry, The Eq. 15 and 16 of the manuscript do require that angular momentum remains a good quantum number. This does not detract from the result, but should be mentioned explicitly. Eq. 15 also suggests that, in a full treatment, the N-invariant characterizes some topological property of the bulk magnetoplasmon band structure. It would be good if the authors were able to comment on this point in the final version of the manuscript.

Reviewer 1	
C1	The authors have convincingly addressed my suggestions and questions. The manuscript has considerably improved and in my opinion it is ready for publication.
R1	We thank the reviewer for the kind response and the recommendation for publication. The quality of the manuscript was increased substantially from their comments.

Reviewer 2	
C1	In their revised version, the authors have added several new details and discussions to help address my earlier concerns. In particular, the proof of nondegeneracy of the susceptibility tensor using the f-sum rule, and the relationship of the N-invariant to the magnetoplasmon angular momentum texture are particularly convincing and illuminating. I am pleased to say that I am now willing to recommend this manuscript for publication in Nature Physics.
R1	We would like to thank the reviewer for their numerous insightful comments and the recommendation for publication. Their suggestions helped increase the quality of the manuscript considerably.
C2	I have one remaining concern, related to the use of rotational symmetry in the derivation of the skyrmion angular momentum. Although the quantization of the N-invariant does not require rotational symmetry, The Eq. 15 and 16 of the manuscript do require that angular momentum remains a good quantum number. This does not detract from the result, but should be mentioned explicitly.
R2	Thank you for bringing this to our attention. We agree that this angular momentum interpretation is limited to the case of rotational symmetry and should be put into context. We have included the following statement in section 4a for clarification: “It should be noted that Eq. (16) only holds in the presence of rotational symmetry, nonetheless the quantization of N remains robust even without symmetry [Eq. (11)].”
C3	Eq. 15 also suggests that, in a full treatment, the N-invariant characterizes some topological property of the bulk magnetoplasmon band structure. It would be good if the authors were able to comment on this point in the final version of the manuscript.
R3	Yes, this is essentially the interpretation of the N-invariant. In principle, the topology of any dipole-carrying excitation is characterized by the N-invariant since they all couple to the electromagnetic field through χ. These can be plasmons but also excitons, phonons, magnons, Cooper-pairs and so on. Our Green’s function formulation treats all these excitations simultaneously. We agree that this point should be clarified better in the text. We have added the following sentences to section 3 to explain the interpretation of the N-invariant in comparison with the TKNN invariant: “Accordingly, the Chern number C characterizes the topology of the bulk electronic band structure while the N-invariant characterizes the topology of the bulk polaritonic band structure. These are plasmons for the quantum Hall fluid, but all dipole-carrying excitations such as excitons, phonons, magnons and Cooper-pairs can be accounted for in our theory.”